# Mutational Status of *SMAD4* and *FBXW7* Affects Clinical Outcome in *TP53*–Mutated Metastatic Colorectal Cancer

**DOI:** 10.3390/cancers14235921

**Published:** 2022-11-30

**Authors:** Sara Lahoz, Adela Rodríguez, Laia Fernández, Teresa Gorría, Reinaldo Moreno, Francis Esposito, Helena Oliveres, Santiago Albiol, Tamara Saurí, David Pesantez, Gisela Riu, Miriam Cuatrecasas, Pedro Jares, Leire Pedrosa, Estela Pineda, Antonio Postigo, Antoni Castells, Aleix Prat, Joan Maurel, Jordi Camps

**Affiliations:** 1Gastrointestinal and Pancreatic Oncology Team, Institut D’Investigacions Biomèdiques August Pi i Sunyer (IDIBAPS), Hospital Clínic of Barcelona, Consorcio de Investigación Biomédica En Red de Enfermedades Hepáticas y Digestivas (CIBEREHD), University of Barcelona, 08036 Barcelona, Spain; 2Translational Genomics and Targeted Therapeutics in Solid Tumors Group, Medical Oncology Department, Hospital Clínic of Barcelona, IDIBAPS, University of Barcelona, 08036 Barcelona, Spain; 3Pharmacology Department, Hospital Clínic of Barcelona, 08036 Barcelona, Spain; 4Pathology Department, Centro de Diagnóstico Biomédico, Molecular Biology CORE, Hospital Clínic, Tumor Bank–Biobank, IDIBAPS, University of Barcelona, 08036 Barcelona, Spain; 5Group of Transcriptional Regulation of Gene Expression, IDIBAPS, Department of Biomedicine, School of Medicine, University of Barcelona, 08036 Barcelona, Spain; 6ICREA, 08010 Barcelona, Spain; 7Department of Cell Biology, Physiology and Immunology, Faculty of Medicine, University Autonomous of Barcelona, 08193 Bellaterra, Spain

**Keywords:** metastatic colorectal cancer, next–generation sequencing, mutational profiling, prognosis, left–sided colorectal cancer, machine learning

## Abstract

**Simple Summary:**

The mutational status of certain genes can be useful to advance therapeutic decision making and clinical management of cancer patients. In metastatic colorectal cancer, current clinicopathological factors employed in clinical practice have low or modest individual effect on survival, leading to a poor ability to discriminate patients at high risk. Here, we obtained data from metastatic colorectal cancer patients undergoing molecular testing by targeted gene sequencing, and we identified *SMAD4* and *FBXW7* mutated genes as negative prognostic markers in *TP53*–driven tumors, which also improved the predictive performance to discriminate high–risk patients beyond clinical factors alone. This negative prognostic impact of co–occurring *SMAD4*/*TP53* and *FBXW7*/*TP53* mutations was confirmed in an independent validation analysis using publicly available data.

**Abstract:**

Next–generation sequencing (NGS) provides a molecular rationale to inform prognostic stratification and to guide personalized treatment in cancer patients. Here, we determined the prognostic and predictive value of actionable mutated genes in metastatic colorectal cancer (mCRC). Among a total of 294 mCRC tumors examined by targeted NGS, 200 of them derived from patients treated with first–line chemotherapy plus/minus monoclonal antibodies were included in prognostic analyses. Discriminative performance was assessed by time–dependent estimates of the area under the curve (AUC). The most recurrently mutated genes were *TP53* (64%), *KRAS* or *NRAS* (49%), *PIK3CA* (15%), *SMAD4* (14%), *BRAF* (13%), and *FBXW7* (9.5%). Mutations in *FBXW7* correlated with worse OS rates (*p* = 0.036; HR, 2.24) independently of clinical factors. Concurrent mutations in *TP53* and *FBXW7* were associated with increased risk of death (*p* = 0.02; HR, 3.31) as well as double–mutated *TP53* and *SMAD4* (*p* = 0.03; HR, 2.91). Analysis of the MSK–IMPACT mCRC cohort (N = 1095 patients) confirmed the same prognostic trend for the previously identified mutated genes. Addition of the mutational status of these genes upon clinical factors resulted in a time–dependent AUC of 87%. Gene set enrichment analysis revealed specific molecular pathways associated with *SMAD4* and *FBXW7* mutations in *TP53*–defficient tumors. Conclusively, *SMAD4* and *FBXW7* mutations in *TP53*–altered tumors were predictive of a negative prognostic outcome in mCRC patients treated with first–line regimens.

## 1. Introduction

Colorectal cancer (CRC) accounts for 10% of all cancer incidence and mortality worldwide, and associated deaths often occur as a consequence of disease metastases [1]. Despite the incremental clinical benefit added by combinatorial targeted therapies over the last two decades, global median survival at five years remains below 15% in patients with disseminated disease [2]. Current clinicopathological factors employed for management of metastatic CRC (mCRC), such as patient performance status, tumor stage at diagnosis, number and diameter of liver metastases, and carcinoembryonic antigen levels, have low or moderate individual effect on survival [3,4,5]. Therefore, further biomarker–driven research on the genomic characterization of mCRC might advance our ability to predict clinical and therapeutic outcomes.

Tumor profiling based on next–generation sequencing (NGS) can provide a molecular rationale for effective prognostication and therapeutic decision making based on individual genomic information. Nevertheless, current guidelines for risk stratification of patients with mCRC are only partially based on the genomic profiling of the tumor [6,7]. Mismatch repair (MMR) deficiency and *BRAF*^V600E^ mutation are relevant genetic markers associated with unfavorable prognosis although their prevalence in mCRC is low [8,9]. In addition, mutations in *KRAS* and *NRAS* have been related to poor survival as well as to lack of response to anti–EGFR therapy [10,11,12]. Other somatically mutated genes reported to hold negative prognostic value affect *SMAD4* [13,14,15] and *FBXW7* in resected liver metastases [16], whose frequency of alteration increases upon advanced CRC stages. Recently, wild–type *APC* in patients with microsatellite–stable (MSS) tumors has been also associated with poor prognosis in mCRC [17,18]. Coexistence of mutations in these frequently altered genes can sometimes exhibit a synergistic effect on survival compared with mutation in one of them alone, such as concurrent triple mutations in *TP53*/*KRAS*/*SMAD4* [19].

In this study, we obtained data from 294 mCRC patients undergoing routine NGS–based tumor testing in our hospital using the Oncomine Solid Tumor 22–gene panel, enabling us to characterize the mutational spectrum of actionable genes in mCRC. We subsequently aimed at assessing the effects on survival of individual and concomitant mutated genes in patients treated with first–line regimens, i.e., chemotherapy doublets plus/minus monoclonal antibodies against EGFR or VEGF. Moreover, we performed predictive modelling using machine learning–based analysis to assess whether genetic biomarkers can improve the diagnostic accuracy over clinical parameters.

## 2. Materials and Methods

### 2.1. Cohort of Patients, Clinical Information, and Ethics Statement

The study was conceived on a retrospective cohort of 294 patients diagnosed with metastatic CRC between January 2016 and January 2020 at the Hospital Clínic of Barcelona. Clinicopathologic information of patients was retrospectively collected from the hospital registry database, and follow–up data were censored in April 2020.

For each patient, disease parameters recorded included sex, age, Eastern Cooperative Oncology Group (ECOG) performance status, stage at diagnosis, site of the primary lesion (ascending or transverse colon were considered right; whilst descending, sigma, or rectum were considered left), microsatellite instability (MSI) status, number and type of affected organs, lactate dehydrogenase (LDH), alkaline phosphatase (ALP), C–reactive protein (PCR), leukocytes, and carcinoembryonic antigen (CEA) blood levels.

Among the total screening population, 200/294 (68%) individuals were treated with approved first–line chemotherapy doublets (FOLFOX, FOLFIRI, or CAPOX) plus/minus anti–VEGF (bevacizumab) or anti–EGFR agents (cetuximab or panitumumab). Only those individuals presenting as good enough medical condition (i.e., ECOG < 2 and/or < 70 years old) were considered suitable for receiving antibody–based targeted agents. Other treatment options were administered to 72/294 (24.5%) patients, including metastatic resection in 50/294 (17%), radiofrequency ablation in 6/294 (2%), and capecitabine monotherapy to 16/294 (6%). The remaining 22/294 (7.5%) patients received best supportive care (BSC) (Appendix A).

### 2.2. Datasets Obtained from Public Repositories

Validation analyses were carried out on publicly available data downloaded from cBioPortal [20]. A cohort of 1134 mCRC patients analyzed with the MSK–IMPACT capture–based NGS of actionable cancer targets [21] was used. Out of the total 1134 cases, 1095 of them (97%) with no missing survival data were included in subsequent prognostic analyses.

Whole–exome sequencing (WES) and RNA sequencing (RNA–Seq) data from the colon and rectal cancer cohorts of The Cancer Genome Atlas (TCGA) (COAD, N = 398; READ, N = 135, respectively) were downloaded from the Genomic Data Commons (GDC) portal on https://gdc.cancer.gov/ (accessed on 29 July 2021). Only primary adenocarcinomas were considered for this study.

### 2.3. Targeted Next–Generation Sequencing

A panel–based NGS strategy was followed at Hospital Clínic of Barcelona using the Oncomine Solid Tumor DNA kit (Thermo Fisher Scientific, Waltham, MA, USA) to interrogate a panel of 22 genes, comprising commonly mutated genes and druggable targets in solid tumors. Tumor DNA was extracted from formalin–fixed paraffin–embedded (FFPE) unstained sections of primary CRC specimens using the QIAmp DNA FFPE kit (Qiagen, Hilden, Germany) and then quantified with Qubit Double–Stranded DNA High Sensitivity assay kit and Qubit fluorometer assay (Thermo Fisher Scientific). Amplicon–enriched libraries were prepared with the IonAmpliSeq Library kit 2.0 (Thermo Fisher Scientific, Waltham, MA, USA) from 10 ng of input DNA per sample using specific primers from the Oncomine Solid Tumor DNA kit (Thermo Fisher Scientific, Waltham, MA, USA). For the massive parallel sequencing of DNA libraries, the Ion Torrent Personal Genome Machine (PGM) platform was used (Thermo Fisher Scientific) according to the manufacturer’s instructions.

Raw data processing and downstream analyses were performed using the Torrent Suite Software version 4.0.2 automated pipeline (Thermo Fisher Scientific), including adapter trimming, quality controls, and alignment to GRCh37/hg19 reference genome. Single–nucleotide variants (SNVs) and short insertions and deletions (Indels) were called and annotated with the Ion Reporter Software (Thermo Fisher Scientific) using the AmpliSeq Colon and Lung Cancer v2 single sample workflow. All detected variants were filtered and prioritized according to a minimal variant allele frequency (VAF) of at least 3%, frequency in normal population < 1% based on ExAC database (SCR_004068), appearance in COSMIC, and pathogenic functional effect as annotated by the ClinVar database (SCR_006169). For downstream analyses, events in KRAS and NRAS were pooled in a single RAS category.

### 2.4. Whole–Exome Sequencing Data Analysis

Mutational profiling to identify tumors harbouring *TP53*, *FBXW7*, and *SMAD4* mutations was performed based on single–nucleotide variants (SNVs) and short insertions and deletions (indels) from WES data. Annotated somatic variants were gathered in aggregated mutation annotation format (MAF). Only common calls detected by both MuTect2 and VarScan2 Somatic Variant Calling Pipelines were encompassed for subsequent analysis. To minimize false–positive events, regions with coverage < 20x and alternative allele frequency <10% were discarded. Truncating alterations (frameshift indels, nonsense and those affecting splicing sites) were directly included as pathogenic. As regards to functional filtering, missense variants were considered pathogenic by the next two available pathogenicity tools: SIFT > 0.05 and/or PolyPhen2 > 0.90. In addition to fulfilling the previous criteria, calls were manually curated and prioritized for final analysis if appearing in COSMIC with FATHMM prediction > 0.70.

### 2.5. RNA–Seq Data Analysis

For transcriptomic analysis, mapped gene read counts from aligned RNA–Seq data (HTSeq–Counts) were analyzed. Gene counts were firstly processed and filtered using *filterByExpr* function (edgeR library) to discard low–level counts. Library size normalization factors were calculated per sample with TMM method by *calcNormFactors* (edgeR) to remove non–biological variability. Filtered data counts were then normalized and transformed into log2–counts per million (logCPM) using *voom* (limma). Gene–level differential expression between sample groups was executed using linear models by *lmFit* and *eBayes* functions (limma), estimating for each gene the log fold–change (FC), Benjamini–Hochberg–adjusted *p*–value (or false–discovery rate, FDR), and the Bayes moderated *t*–statistics. Differentially expressed genes were considered when FC > |1.5| and FDR < 0.15. In order to identify relevant biological processes related to deregulated genes implicated in patient phenotypes (i.e., mutated vs. wild–type tumors), gene set enrichment analysis (GSEA) was conducted on the GSEA desktop application 4.1.0 (Broad Institute, Cambridge, MA, USA). Gene sets or pathways were judged as enriched with a minimum normalized enrichment score (NES) > |1.75| and *p*–value < 0.05.

### 2.6. Study Endpoints and Statistical Analysis

Two endpoints were evaluated for survival analysis: the primary objective was overall survival (OS), defined as the time from metastatic diagnosis to the date of death by any cause (in deceased patients) or last follow–up. The secondary objective was progression–free survival (PFS), defined from the date of diagnosis of metastasis to the date of progression or death (in patients with event) or last visit. A third study endpoint included in the analysis was objective response rate, as defined in RECIST v1.1 clinical guidelines [22].

The Kaplan–Meier method was used to estimate time–related events, and a log–rank test at 5% significance level was used to compare OS/PFS between groups. To calculate hazard ratios (HRs) and corresponding 95% confidence intervals (CIs), univariate and multivariable Cox regression models with proportional hazards were fitted to assess risks of each mutated gene (individually or in combination) on OS/PFS, adjusting for baseline clinical variables (sex, age, ECOG performance status, MSI, number of affected organs, site of primary lesion, LDH, ALP, PCR, and leukocytes blood levels). Only those clinical variables showing a *p*–value < 0.15 at the univariate setting were included as co–variates for multivariable Cox models. Multivariate logistic regression was used to judge the association of biomarkers with the degree of response, and odds ratios (OR) were derived as the exponential of each beta coefficient obtained from regression equations. All tests were two–sided, and the significance threshold of all *p*–values was set at the 5% level. All statistics were carried out using R software version 3.6.

### 2.7. Machine Learning–Based Classifier Model

Predictive modelling was computed integrating all covariates (including clinical features plus/minus the mutational status of selected genes) to evaluate the probability of each patient to experience disease progression or death. Multiple imputations with chained equations were run using package *mice* to replace missing data by imputed values, thereby avoiding subsequent exclusion of these cases by the classifier algorithm. A gradient boosting machine (package *gbm* v. 2.1.8) algorithm was employed for patient classification and for computing individual probabilities. Feature selection was performed on each training dataset by using a Least Absolute Shrinkage Selector Operator (LASSO) regression from R–package *glmnet* version 2.0–18 (SCR_015505) so that only those parameters with a Beta–coefficient > 0 in the LASSO equation were subsequently included in the gradient boosting machine–based classifier. The model was trained using 5–fold cross–validation with a training/testing split ratio of 80:20% including a repeated random sub–sampling procedure to ensure statistical robustness. Diagnostic accuracy of machine learning–based classifications was evaluated in terms of sensitivity, specificity, precision, and accuracy calculated with *crossval* package. Discriminative ability of the biomarkers was assessed by means of the area under the curve (AUC) based on time–dependent receiver operating characteristic (ROC) curve analysis, computed using *timeROC* package.

## 3. Results

### 3.1. Clinico–Pathological Description of the Study Population

Table 1 summarizes demographic and clinico–pathological characteristics of the final study cohort. Only those patients receiving baseline chemotherapy doublets as first–line treatment (N = 200) were included for survival analysis to minimize prognostic biases related to therapy (Appendix A). Median age at diagnosis was 66 years (range 30–85), and 57.5% of individuals were male. Four patients (2%) had tumors with microsatellite instability (MSI), and 148/200 (74%) had primary tumors located on left side. Clinico–pathological features according to tumor location are described in Appendix A. Median follow–up time was 19.90 months (range 2–97.2 months). At the time of censoring, 100/200 patients (50%) had died and 156/200 (78%) had progressed to first–line treatment, of whom 109 subsequently underwent a second–line therapeutic option.

To evaluate the association between baseline clinical parameters and OS, Cox proportional hazards models were fitted comprising the study population (N = 200) (Table 2). The strongest independent predictor for OS was the Eastern Cooperative Oncology Group (ECOG) performance status (*p* < 0.0001; HR = 2.28; 95% CI: 1.56–3.33) as a continuous variable. Other clinicopathological parameters associated with shorter OS at the multivariable setting were lactate dehydrogenase (LDH) (*p* = 0.0009), alkaline phosphatase (ALP) (*p* = 0.005), leukocytes (*p* = 0.005), and age (*p* = 0.015) as continuous measures. Moreover, patients with left–sided tumors exhibited longer mean OS as compared to those with right–sided ones (22 vs. 16 months, respectively, *p* = 0.025 in a univariate analysis) (Table 2, Appendix A).

### 3.2. Recurrently Mutated Genes in mCRC

Multi–gene profiling of the entire 294–patient study cohort revealed a total of 526 somatic mutation events, comprising single–nucleotide variants (SNVs) and short insertions and deletions (Indels). Overall, 268/294 (91%) tumors carried one or more genes mutated (Figure 1). The most frequently mutated genes were *TP53* (188 patients, 63.95%), *RAS* (145 patients, 49.32%), *PIK3CA* (45 patients, 15.31%), *SMAD4* (40 patients, 13.61%), *BRAF* (39 patients, 13.27%), and *FBXW7* (28 patients, 9.52%). The remaining genes appeared mutated in less than 5% of cases. As expected, over 90% of mutations affecting *RAS* were missense variants, and 83% of mutations in *KRAS* altered hotspot codons 12 or 13. Mutations in *RAS* positively correlated with occurrence of *PIK3CA* events (*p* = 0.015). Tumors with MSI (12 patients, 4.1%) were enriched for *BRAF* mutations (*p* = 0.0002), showed inferior proportion of *TP53* and *RAS* events (*p* = 0.01 and *p* = 0.035, respectively), and were more frequent in the right than left colon (10% vs. 1.5%, respectively; *p* = 0.002) as compared to MSS tumors. In addition, *FBXW7* was more frequently mutated in left–sided than right–sided colorectal tumors (11.1% vs. 6.4%, respectively; *p* = 0.38) (Appendix A). Although mutational frequencies did not differ significantly among treatment groups, *BRAF* and *PIK3CA* mutations were enriched in patients treated with best supportive care (BSC). Finally, rectal, descendent colon, and sigmoid cancers were pooled together following their similar mutation frequencies (Appendix A) except for *RAS* and *FBXW7*, both appearing significantly more mutated in rectal than in left–sided tumors (Fisher’s *p* = 0.02 and *p* = 0.005, respectively).

### 3.3. Prognostic Estimates of PFS and OS by Individual Mutated Genes

We subsequently assessed the independent prognostic value of each mutated gene using Cox statistics in the population of 200 patients treated with first–line regimens (Table 3). In a univariate analysis, *PIK3CA* mutations were predictive of PFS and OS (*p* = 0.01 and *p* = 0.007, respectively). At the multivariable setting, mutations in *FBXW7* correlated with worse OS rates (*p* = 0.036) and mutations in *SMAD4* predicted PFS (*p* = 0.0015). Patients with *BRAF*–mutant tumors exhibited shorter median OS times compared to their wild–type counterparts though this difference was not significant (14 vs. 29 months, *p* = 0.39). Among left–sided tumors, *FBXW7* mutational status emerged as an independent predictor of both PFS and OS (*p* = 0.05 and *p* = 0.03, respectively), while *SMAD4* predicted PFS only (*p* = 0.0075) when adjusting for clinical variables (Appendix A). In contrast, *RAS* mutations moderately predicted worse PFS independently of clinical features in the right colon (*p* = 0.044).

### 3.4. Survival Modelling Based on the Combination of Mutated Genes

We next sought to investigate to what extent co–occurrence of mutations might impact patient prognosis (Table 3). Among tumors harboring baseline *TP53* mutations, concomitant variants in *SMAD4* (17/200 patients, 8.5%) independently correlated with reduced PFS (*p* = 0.0002; HR, 4.32; 95% CI, 2–9.30) and OS (*p* = 0.035; HR, 2.91; 95% CI, 1.08–7.85) at the multivariable setting (Figure 2A). Median time until progression was 6 months for *TP53*–mutant cases with co–mutated *SMAD4* versus 9.8 months in the *SMAD4* wild–type group; while median time until decease was 26 months versus 28.5 months for the abovementioned groups, respectively. As regards to patients with *TP53*–mutant tumors with concomitant altered *FBXW7* (16/200 patients, 8%), they were at a significantly increased risk of progression (*p* = 0.025; HR, 2.65; 95% CI, 1.13–6.23) and death (*p* = 0.019; HR, 3.31; 95% CI, 1.22–8.96) at the multivariable setting (Figure 2B). Median time until progression was 4.9 months in those high–risk cases harboring *TP53*/*FBXW7* double mutations versus 10 months in low–risk cases with single mutations in *TP53*; and median time until decease was 23.7 months versus 29.2 months, respectively. When considering left–sided tumors exclusively (N = 100, 75.8%) within the *TP53*–defficient population, concurrent *TP53*/*FBXW7* mutations exhibited independent ability to estimate PFS (*p* = 0.02; HR, 2.80; 95% CI, 1.14–6.89) and OS (*p* = 0.02; HR, 3.65; 95% CI, 1.20–11.06), while *TP53/SMAD4*–mutated patients exhibited inferior PFS only (*p* = 0.002). Among individuals with right–sided tumors (N = 32, 24.2%), concurrent mutations affecting *TP53*/*SMAD4* conferred significantly reduced PFS (*p* = 0.026) and OS (*p* = 0.003) (Appendix A).

The association of mutations in *SMAD4* and *FBXW7* with treatment response was assessed by multivariate logistic regression, excluding the 24 non–evaluable response cases due to death or loss of follow–up prior to response evaluation. Among patients with mutations in *SMAD4*, only one case (4%) achieved complete response (CR), seven cases (27%) achieved a partial response, while eighteen (69%) were non–responsive, i.e., presented stable disease or progressed (Fisher’s *p* = 0.036). Accordingly, patients with *SMAD4* mutations displayed a trend towards poor response rates (*p* = 0.106; OR, 1.48; 95% CI, 0.92–2.38), which was also observed for double–mutant *TP53*/*SMAD4* cases (*p* = 0.175; OR, 1.49; 95% CI, 0.84–2.64) (Appendix A). Among patients with *FBXW7* or *TP53*/*FBXW7* mutated tumors, none of these mutations appeared to be significantly associated with response rates.

Intended for a validation analysis, public data from 1095 mCRC patients sequenced with the MSK–IMPACT gene–panel were downloaded from cBioPortal [20]. We modeled the reported oncogenic DNA mutations for their relationship with OS, correcting for the available clinical variables (i.e., age, sex, primary tumor location, and microsatellite status). In this independent dataset, patients with double mutations affecting both *TP53*/*SMAD4* and *TP53*/*FBXW7* showed a trend towards reduced OS rates at the multivariable setting (*p* = 0.13 and *p* = 0.096, respectively) (Appendix A). In this sense, median time until decease was 26 months for *TP53*/*SMAD4* double–mutant patients versus 28.5 months for *TP53* single–mutated cases. Concerning those individuals with *TP53*/*FBXW7* mutant tumors, their median time until death was 23.7 months compared with 29.2 months for those with individually mutated *TP53*.

### 3.5. Gene Expression Profiling Underlying Double–Mutation Genotypes for TP53/SMAD4 and TP53/FBXW7

To evaluate potentially deregulated biological processes underlying *SMAD4* and *FBXW7* mutations in the *TP53*–defficient CRC population, we analyzed WES and RNA–seq data from 533 colorectal tumors conforming the TCGA cohorts COAD and READ. Within the *TP53*–mutated population (N = 251/533, 47.1%), mutational frequencies for *SMAD4* and *FBXW7* were 11.6% and 13.1%, respectively. Differential expression analysis comparing tumors being *TP53/SMAD4* double–mutant versus *TP53* single–mutated revealed 28 significantly upregulated genes, comprising *SOX2*, *TM4SF4*, and *CALB1* (log2 FC > 2, *p* < 0.01), and 59 significantly downregulated, including *LY6G6D*, *SPACA3*, *CEACAM7*, and *MS4A12* (log2 FC < −2, *p* < 0.05) (Figure 3A). GSEA confirmed the downregulation of the TGF–beta receptor signaling pathway as the main downregulated process (NES = −2.02, *p* = 0.004) in tumors with *SMAD4* mutation. In cases having double–mutated *TP53*/*SMAD4*, the most downregulated gene sets (NES < −2.00, FDR < 0.25) were the transforming growth factor–β–activated kinase 1 (TAK1) activity, JNK–dependent phosphorylation of c–Jun, and activation of IKKs complex by TAK1 (Figure 3B,C).

When comparing tumors bearing a *TP53*/*FBXW7* double–mutated genotype with those being *TP53* mutated alone, differential gene expression analysis indicated 287 significantly upregulated genes, including *MAGEA3*, *MAGEA12*, *MAGEA6*, *CALB1*, and *NPSR1* (log2 FC > 2.75, *p* < 0.001), and 582 downregulated genes, encompassing *ACSL6*, *WIF1*, *LY6G6D*, *SLC13A2*, *F7*, and *NALF2* (log2 FC < −1.75, *p* < 0.01) (Figure 3D). For those tumors having a *TP53*/*FBXW7* double–mutated genotype, the most significantly enriched cancer–related signaling pathways based on a NES > |1.75| and nominal *p* <  0.05 were the upregulation of glycolysis and metabolism of folate and pterines, overexpression of TP53 activity–related signaling, defective intrinsic pathway of apoptosis, interleukin 12 family signaling, CD28–dependent PI3K/AKT activity, homologous recombination, and alteration of the Fanconi anemia pathway (Figure 3E,F). Intriguingly, the former two gene sets were only significantly upregulated in *FBXW7* mutant tumors within the *TP53*–deficient population but not in the whole population.

### 3.6. Discriminative Performance of the Clinico–Genetic Biomarkers

Prognostic–based estimates of the AUC were calculated in a time–dependent manner, aiming at examining the discriminatory ability of clinical parameters along with the mutational status of the most frequently altered genes. Three distinct biomarker compositions were defined: clinical–based, genetic–based, and clinico–genetic (mixed model) (Appendix A). The entire set of risk predictors was composed by the mutational status of *TP53*, *SMAD4,* and *FBXW7*, plus the clinical variables age, sex, ECOG performance status, number of affected organs (one or more), MSI status, location of the primary tumor (right or left), LDH, ALP, leukocytes in blood, and PCR levels.

Both the clinico–genetic and clinical–based models displayed the highest AUC for death at the 1000–day time–point, while the genetic–based exhibited the strongest discriminative ability at 400 days (Figure 4A). As expected for predicting disease progression, the highest AUC values were achieved earlier in time than for death (Figure 4B). In the study cohort, median times until progression and death were 9.5 and 14 months, respectively, which were the time–points subsequently used to infer the optimal time–dependent AUCs. The clinico–genetic model ranked an optimal AUC for death of 87% (95% CI, 81–92%) at the 9.5–month time–point and for progression of 77% (95% CI, 70–84%) at the 14–month time–point; the clinical–based, an AUC for death of 85% (95% CI, 79–91%) and for progression of 0.74 (95% CI, 0.66–0.81); and the genetic–based, an AUC for death of 55% (95% CI, 46–65%) and for progression of 55% (95% CI, 46–63%) (Figure 4).

When applied to the entire clinico–genetic biomarker model, LASSO selected a median number of 12/16 (75%) parameters per fold, ranging from 5 to 13 (Appendix A). When considering only clinical parameters, the patients’ age, ECOG performance status and MSI status were chosen in 100% of folds, while sex and number of affected organs in 80%, and PCR levels in 60%. Among the analysed mutated genes, *SMAD4* and *PIK3CA* were chosen in 100% of folds while *FBXW7* was included in 40% of folds (Appendix A). As regards to the efficacy of predictions, the machine–learning model including the entire set of clinical plus mutational parameters achieved 79% sensitivity and 71% specificity to predict patient death status, as opposed to a 74% sensitivity and 72% specificity for the model only containing clinical variables (Appendix A).

## 4. Discussion

Personalized medicine requires accurate understanding of individual tumor biology for effective prognostication and prediction of clinical outcomes, ultimately intending to guide therapeutic interventionism. The implementation of molecular screening programs using NGS–based assays has been already addressed at major cancer centers [6,21,23]. In the present study, we consecutively collected data from 294 mCRC cases that underwent routine mutational profiling at our institution between January 2016 and January 2020 using the Oncomine Solid Tumor panel, which comprises the most frequently mutated and actionable genes in CRC except for *APC*. Our results revealed a negative prognostic effect of *SMAD4* and *FBXW7* mutations in patients with *TP53*–mutated tumors, which was partly confirmed in a separate larger cohort of mCRC cases. The addition of these mutated genes upon clinical factors caused an improvement in the discriminative ability to predict progression and death. Despite the retrospective single–institution design of our study, we demonstrate the feasibility of routinely implemented genetic testing to improve patient stratification for treatment and prognostic procedures in mCRC.

A substantial amount of evidence has already shown that the loss of the tumor suppressor *SMAD4* correlates with metastasis occurrence and poor chemo–response in patients with CRC [24,25]. Inactivating *SMAD4* mutations occur in approximately 10 to 16% of colorectal tumors [26], similar to the 14% frequency of mutated cases appearing in our cohort. Moreover, inactivation of *SMAD4* can also be mediated by loss of heterozygosity at chromosome locus 18q21, exhibited by approximately 50 to 66% of CRCs [27,28]. Reduced levels of SMAD4 are known to correlate with disease recurrence and shorter survival in mCRC patients [13,14,29], consistent with our findings regarding the negative prognostic effect of either sole *SMAD4* mutations or combined with mutated *TP53*. According to our analysis, *SMAD4* mutations confer superior risk of progression and death when co–occurring with mutant *TP53* than when occurring alone or combined with *RAS* mutations, potentially as a consequence of a dysregulated activity of the IKK complex and the JNK–dependent phosphorylation of c–Jun and PTK6, which appear to be altered in *SMAD4* mutant tumors when present in the *TP53*–altered population exclusively. In line with recent findings [19], the 11 patients in our cohort with triple–mutant tumors for *TP53*/*RAS*/*SMAD4* exhibited reduced survival compared to those with mutated *RAS* or *TP53* alone. Nonetheless, this triple combination did not show greater prognostic value than *TP53*/*SMAD4* double mutation, suggesting a synergistic functional interaction between SMAD4 and p53 loss [30]. Our results suggest that patients with *SMAD4*–mutated tumors encompass reduced rates of treatment response, supporting several studies that point out *SMAD4* inactivation as predictive marker for chemo–resistance to 5–fluoroacil (5FU) –based chemotherapy [29,31,32]. Furthermore, mutant *BRAF* is also a well–known predictor of prognosis in mCRC [33]. In our cohort, the number of *BRAF*–mutated individuals was relatively small (N = 23), probably explaining the lack of statistical significance of Cox hazards models; even though, patients with *BRAF*–mutant tumors displayed evident shorter median OS times compared to wild–types. For mutated *KRAS*, although it has also been reported to correlate with poor outcome in advanced CRC [34], in our set it was only prognostic for PFS at the univariate setting in right–sided tumors. Additionally, recent reports showed that tumors lacking *APC* mutations correlate with worse prognosis [17,18], in agreement with results we obtained from the MSK–IMPACT dataset (data not shown).

Less is known about the role of *FBXW7* as a prognosis marker in mCRC. *FBXW7* is a tumor suppressor gene encoding a subunit of the Skip1–Cull–F–box (SCF) ubiquitin ligase complex, which controls proteasome–mediated degradation of various cell cycle regulators such as cyclin E, c–Myc, Notch1, and mTOR [35]. Furthermore, *FBXW7* also regulates early apoptosis [36], stem cell differentiation [37,38], and chromosomal stability through p53–related signaling [39,40]. Monoallelic mutations affecting *FBXW7* have been frequently detected in several cancer types, including sporadic cholangiocarcinoma (35%), T–cell acute lymphocytic leukemia (31%), endometrial carcinoma (16%), and CRC (16%), among others [41,42]. In our cohort, 23 patients (11.5%) were carriers of *FBXW7*–mutated tumors. A novel finding is that co–occurrence of mutations in *FBXW7* and *TP53* in 8% of the cases holds prognostic value in mCRC. Coexistent variants in these genes have been previously documented in gastric cancer [43] and more recently in breast cancer, where the authors suggested that the two mutations cooperate for breast tumorigenesis [44]. In CRC in vitro models, the reduction of phospho–p53(Ser15) has been suggested to contribute to decrease the therapeutic efficacy of oxaliplatin in *FBXW7*–mutated cells [45]. Our transcriptomic analysis of *TP53*/*FBXW7* double–mutated tumors pointed to the dysregulation of pathways related to P53 and DNA damage response (DDR). Although knowledge is still limited, several studies already suggested that *FBXW7* is involved in controlling genomic stability by targeting a wide range of DDR regulatory proteins such as P53 [46,47], PLK1 [48], and SOX9 [49]. Clinical significance of *FBXW7* inactivating alterations in CRC remains elusive although it is one of the most frequently mutated genes in metastic disease [50]. Loss of *FBXW7*, located at chromosome 4q32, correlates with disease progression and usually appears as a late event during the colorectal tumorigenesis [51]. As reported previously, *FBXW7* missense mutations have been also associated with shorter OS in mCRC [16] and in patients undergoing resection of colorectal liver metastases [52]. Conversely, some other authors reported that *FBXW7* mutations do not confer differences in disease–free survival [53]. Our results further support that patients with *FBXW7*–mutated tumors show poorer OS, even at the multivariable setting when including determining clinical variables. Remarkably, mCRC patients with concomitant *FBXW7*/*TP53* tumor mutations showed not only lower OS but also poorer PFS. To be mentioned, although the percentage of patients receiving anti–EGFR agents was inferior in those *TP53*/*SMAD4* and *TP53*/*FBXW7* double–mutated cases as compared to their wild–type counterparts, this difference did not appear to be statistically significant (Fisher’s *p* = 0.49 and *p* = 0.55 for *TP53*/*SMAD4* and *TP53*/*FBXW7* groups, respectively). Additionally, mutations in *FBXW7* resulted in a more intense effect on OS than PFS, in contrast to *SMAD4*–mutated tumors, as it is illustrated by the violation of the Cox proportional hazards showed by the *FBXW7* Kaplan–Meier curve in the MSK–IMPACT cohort. Moreover, the analysis of our cohort showed that *FBXW7* mutations might be prognostic specifically in the set of patients treated with first–line cetuximab or panitumumab, unlike the MSK–IMPACT dataset, which was composed by treatment–disparate patient groups. To note, in our cohort a total of 18 (78%) tumors with mutated *FBXW7* were located on the left side, reinforcing the idea that *FBXW7* mutations might act as biomarkers of poor outcome associated to left–sidedness, which often exhibits a moderately favorable prognosis compared to right–sided colon mCRC [54]. Furthermore, *FBXW7* mutations have been reported to be prevalent in patients non–responsive to anti–EGFR treatment [55], which supports that *FBXW7* inactivation in *RAS* wild–type tumors confers resistance to cetuximab or panitumumab in mCRC patients [56]. In our cohort, we found no significant association of *FBXW7*, *SMAD4*, *TP53*/*FBXW7*, or *TP53*/*SMAD4* mutational status with treatment response, possibly due to the low number of patients included in the mutated groups.

A number of studies have reported the limited clinical significance of single tumor markers in terms of discriminative performance [57,58]. In our multivariable prognostic models, we included a plethora of clinicopathological parameters routinely collected and registered at our institution for patient clinical management. However, none of them resulted in a significant increase in the discriminatory accuracy except for the ECOG PS, which added a 10–12% on the multivariable AUC value and was selected in all cross–validation folds along with MSI status. Likewise, despite the prognostic strength of our identified mutated genes, their added predictive value was limited, resulting in only a 2–3% increase in the time–dependent AUC score and a 5% incremental sensitivity to predict patient overall death status, which is similar to other studies assessing the incremental diagnostic ability by specific mutated genes [57,59]. These results emphasize the importance of including strong prognostic clinical co–variates in multivariable predictive models in mCRC.

## 5. Conclusions

This study highlights a negative prognostic effect of *SMAD4* and *FBXW7* mutations in mCRC patients with *TP53*–driven tumors when receiving first–line chemotherapy doublets/minus anti–EGFR or anti–VEGF. Hence, the mutational analysis of these three genes alongside baseline clinical parameters might serve as useful biomarkers to select those mCRC patients at high risk of progression or death for therapeutic decision making.

## Figures and Tables

**Figure 1 cancers-14-05921-f001:**
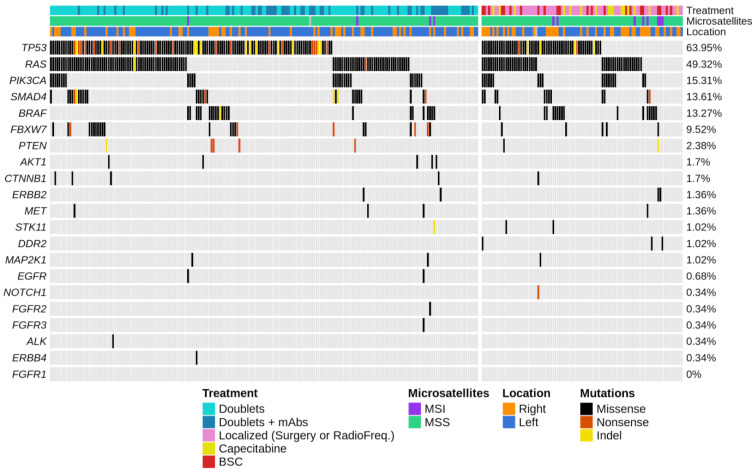
**Mutation profiling of actionable and cancer driver genes interrogated by targeted capture–based sequencing in patients with metastatic colorectal cancer.** Oncoprint chart illustrating the mutation landscape of the Oncomine Solid Tumor 22–gene panel, ranked by ordered mutational frequencies (right), for the 294 capture–sequenced colorectal tumors arranged in two groups according to treatment received (i.e., left group comprises N = 200 patients treated with chemotherapy doublets plus/minus monoclonal antibodies, mAbs; right group comprises N = 94 patients treated with other second-line options or BSC). Each row represents a gene and each column an individual tumor sample. Colored bar plots (top) provide data on tumor–related and treatment characteristics annotated in legend (bottom).

**Figure 2 cancers-14-05921-f002:**
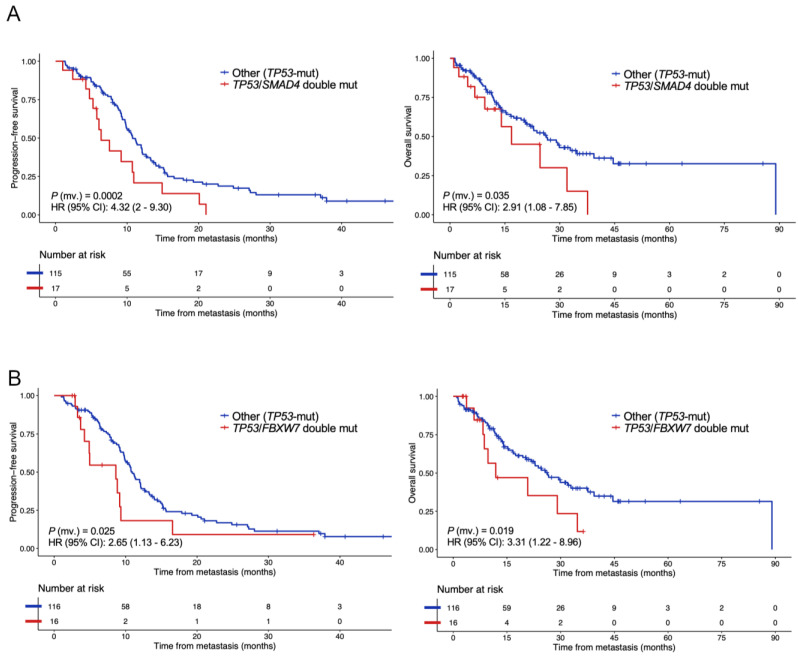
**Prognostic value of double mutations in *TP53*/*SMAD4* and *TP53*/*FBXW7* in metastatic colorectal cancer patients.** Kaplan–Meier estimates displaying the cumulative proportion of individuals (*Y*–axis) who were progression– or death–free over the study period (*X*–axis), and Cox–derived resulting statistics for PFS and OS in the study population (N = 132/200, 66%) analyzed with the Oncomine panel–based sequencing, stratifying by the presence or absence of coexistent mutations in (**A**) *TP53*/*SMAD4* and (**B**) *TP53*/*FBXW7*. *p*–values were obtained using the log–rank test and hazard ratios using a Cox proportional hazards model, correcting for baseline clinical factors (see Methods). mv., multivariable.

**Figure 3 cancers-14-05921-f003:**
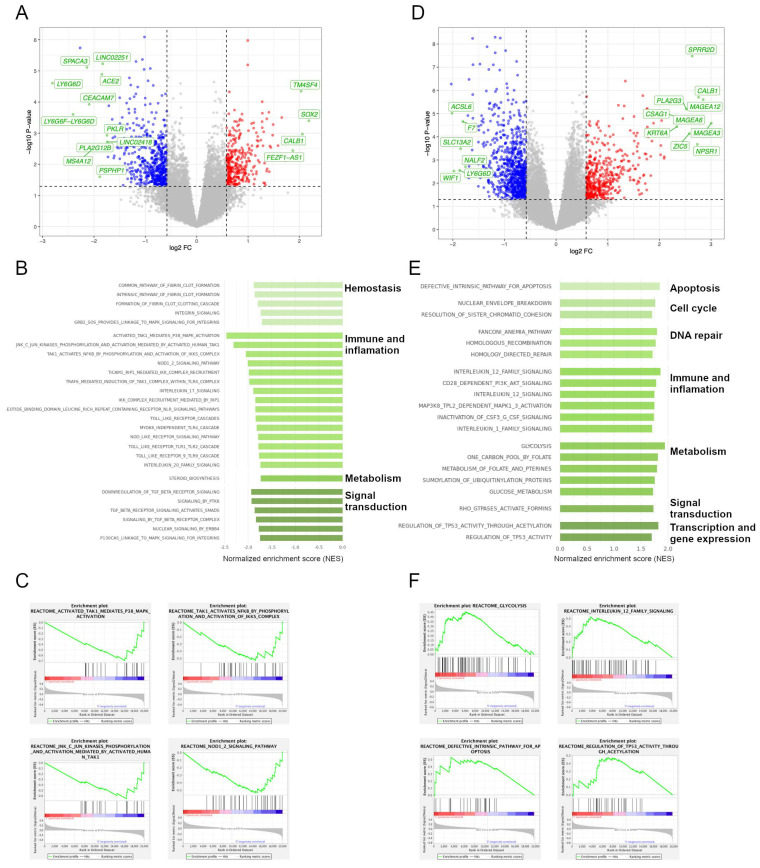
**Deregulated cancer–related transcriptomic pathways underlying *SMAD4* and *FBXW7* mutations in *TP53*–defficient tumors.** (**A**,**B**) Volcano plots displaying differentially expressed genes in tumors harboring a mutated genotype for (**A**) *TP53/SMAD4* and for (**B**) *TP53/FBXW7*. Grey circles indicate genes with non–significant expression differences, red circles indicate significantly overexpressed genes (with log2 FC > 0.58 and non–adjusted *p* < 0.05), and blue circles label significantly under–expressed genes (with log2 FC < −0.58 and non–adjusted *p* < 0.05). Genes that are highlighted in green show the highest FC in gene expression (for *TP53/SMAD4*: log2 FC > 1.5 or log2 FC < −1.5; for *TP53/FBXW7*: log2 FC > 2 or log2 FC < −2). (**C**,**D**) Most significantly enriched gene sets and categories from REACTOME and KEGG in double–mutated (**C**) *TP53/SMAD4* and (**D**) *TP53/FBXW7* tumors, according to a NES > |1.75| and nominal *p* < 0.05. (**E**,**F**) Exemplary GSEA plots of those significant gene sets with the highest enrichment scores in (**E**) *TP53/SMAD4*–mutated and (**F**) *TP53/FBXW7*–mutated cancers.

**Figure 4 cancers-14-05921-f004:**
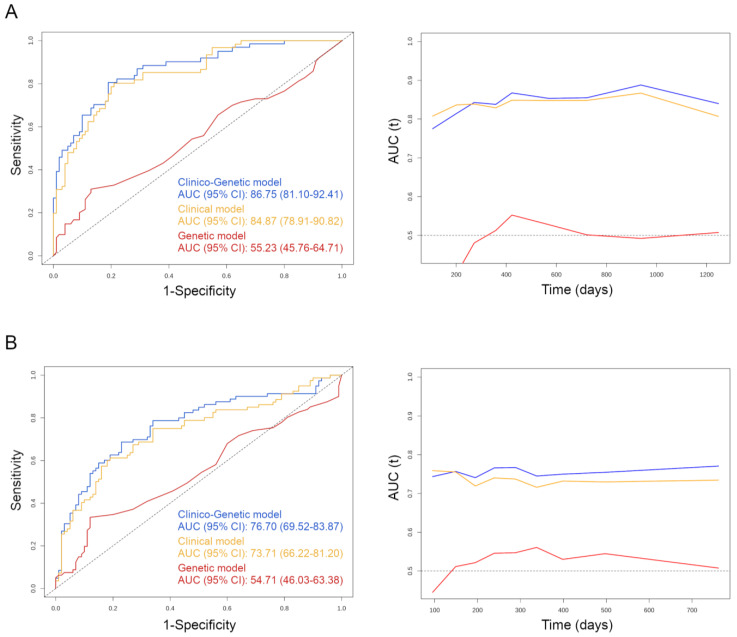
**Discriminatory performance of the three mCRC biomarker models for predicting clinical outcomes of patients.** (**A**) Time–dependent estimates of the AUC (**left**) for prediction of overall survival, visualized on ROC curves for each of the three biomarker combinations, alongside a diagram (**right**) displaying the temporal evolution of the AUC value for the different models. (**B**) Same performance metrics for discerning those patients at risk of progression.

**Table 1 cancers-14-05921-t001:** Clinical description of our study cohort.

Variable	All Patients	Patients Unmutated for *TP53*/*SMAD4* or *TP53*/*FBXW7*	Patients Mutated for *TP53*/*SMAD4*	Patients Mutated for *TP53*/*FBXW7*
Num. of patients	200 (100%)	169/200 (85.5%)	17/200 (8.5%)	16/200 (8%)
Age, years (median, range)	66 (30–85)	65 (30–85)	63 (42–79)	67 (53–77)
Sex (number, %)				
Male	115 (57.5%)	96 (56.8%)	11 (64.7%)	9 (56.25%)
Female	85 (42.5%)	73 (43.2)	6 (35.3%)	7 (43.75%)
Primary tumor location (number, %)				
Right	52 (26%)	46 (27.2%)	4 (23.5%)	2 (12.5%)
Left	148 (74%)	123 (72.8%)	13 (76.5%)	14 (87.5%)
Stage at diagnosis (number, %)				
I	6 (3%)	6 (3.6%)	0 (0%)	0 (0%)
II	13 (6.5%)	13 (7.7%)	0 (0%)	0 (0%)
III	42 (21%)	30 (17.7%)	7 (41.2%)	5 (31.3%)
IV	139 (69.5%)	120 (71%)	10 (58.8%)	11 (68.7%)
Metastatic sites (number of organs affected) (number, %)				
1	93 (46.5%)	76 (45%)	9 (53%)	9 (56.25%)
>1	107 (53.5%)	93 (55%)	8 (47%)	7 (43.75%)
ECOG PS (number, %)				
0	73 (36.5%)	55 (33%)	9 (53%)	10 (62.5%)
1	87 (43.5%)	80 (47%)	5 (29.4%)	3 (18.75%)
>1	38 (19%)	32 (19%)	3 (17.6%)	3 (18.75%)
No data	2 (1%)	2 (1%)	0 (0%)	0 (0%)
LDH, units/L (median, range)	351.5 (99–18,585)	351.5 (99–18,585)	331.5 (152–967)	396.5 (137–2856)
ALP, units/L (median, range)	100 (36–902)	105 (36–902)	109 (65–450)	89 (57–473)
PCR, mg/L (median, range)	2 (0–73.4)	2 (0–73.4)	1.62 (0–14.92)	2.4 (0.04–22.03)
Leukocytes, 10^6^/L (median, range)	7600 (1294–32,290)	7730 (1294–32,290)	7360 (4400–15,470)	7125 (4890–14,580)
CEA, ng/mL (median, range)	19.35 (0.3–7956)	20.75 (0.9–7956)	35.2 (0.3–1274)	10.8 (2.1–1179)
1st line of treatment				
Doublets (FOLFOX/FOLFIRI/CAPOX)	139 (69.5%)	114 (67.5%)	14 (82.3%)	13 (81.25%)
Doublets + anti–EGFR (cetuximab/panitumumab)	41 (20.5%)	39 (23%)	1 (6%)	1 (6.25%)
Doublets + anti–VEGF (bevacizumab)	20 (10%)	16 (9.5%)	2 (11.7%)	2 (12.5%)
MMR status (number, %)				
Proficient	195 (98%)	164 (97.5%)	17 (100%)	16 (100%)
Deficient	4 (2%)	4 (2.5%)	0 (0%)	0 (0%)
Primary tumor resected (number, %)	126 (63%)	104 (61.5%)	12 (70.6%)	10 (58.8%)
Location of metastases (number, %)				
Liver metastasis	149 (74.5%)	127 (75%)	9 (53%)	13 (81.25%)
Lung metastases	65 (32.5%)	55 (32.5%)	6 (35.3%)	4 (24%)
Peritoneal metastases	30 (15%)	23 (13.65%)	6 (35.3%)	1 (6.25%)
Node metastases	68 (34%)	58 (34.3%)	4 (23.4%)	6 (37.5%)

ECOG PS, European Cooperative Oncology Group performance status; LDH, lactate dehydrogenase; ALP, alkaline phosphatase; PCR, protein C reactive; CEA, carcinoembryonic antigen; MMR, mismatch repair.

**Table 2 cancers-14-05921-t002:** Cox models for PFS/OS stratified by the annotated clinical variables.

Variable		PFS				OS			
		Univariate Analysis		Multivariable Analysis		Univariate Analysis		Multivariable Analysis	
		HR (95% CI)	*p*	HR (95% CI)	*p*	HR (95% CI)	*p*	HR (95% CI)	*p*
Age	<continuous>	–	0.11	–	0.43	–	**0.0001**	–	**0.015**
Sex	Female vs. Male	0.85 (0.62–1.17)	0.32	0.70 (0.48–1.02)	0.06	0.91 (0.61–1.35)	0.64	0.71 (0.43–1.16)	0.17
Location of primary tumor	Right vs. Left	1.19 (0.82–1.73)	0.36	1.09 (0.69–1.72)	0.71	1.66 (1.06–2.59)	**0.025**	1.16 (0.67–1.96)	0.57
Metastatic sites (number organs affected)	1 vs. >1	1.88 (1.35–2.61)	**0.0002**	1.97 (1.28–3)	**0.0015**	1.82 (1.21–2.75)	**0.004**	1.21 (0.96–1.53)	0.11
MMR status	MSS vs. MSI	0.31 (0.10–0.99)	0.049	0.42 (0.56–3.21)	0.41	0.32 (0.1–1.03)	0.056	0.21 (0.03–1.61)	0.13
ECOG PS	1/2/3 vs. 0	1.72 (1.23–2.40)	**0.002**	1.30 (0.83–2.05)	0.25	3.75 (2.27–6.2)	**<0.0001**	2.1 (1.07–4.12)	**0.03**
	2/3 vs. 0/1	2.90 (1.98–4.25)	**<0.0001**	2.36 (1.44–3.88)	**0.0007**	6.29 (4.04–9.8)	**<0.0001**	3.39 (1.97–5.84)	**<0.0001**
	<continuous>	1.77 (1.41–2.23)	**<0.0001**	1.57 (1.15–2.14)	**0.0045**	3.41 (2.53–4.6)	**<0.0001**	2.28 (1.56–3.33)	**<0.0001**
LDH	<continuous>	–	**<0.0001**	–	**0.006**	–	**<0.0001**	–	**0.0009**
ALP	<continuous>	–	**<0.0001**	–	0.057	–	**<0.0001**	–	**0.005**
PCR	<continuous>	–	**0.0007**	–	0.47	–	**0.002**	–	0.86
Leukocytes	<continuous>	–	**<0.0001**	–	0.126	–	**<0.0001**	–	**0.008**
CEA	<continuous>	–	**0.017**	–	0.85	–	**0.0002**	–	0.98

PFS, progression–free survival; OS, overall survival; HR, hazard ratio; CI, confidence interval; MMR, mismatch repair; ECOG PS, European Cooperative Oncology Group performance status; LDH, lactate dehydrogenase; ALP, alkaline phosphatase; PCR, protein C reactive; CEA, carcinoembryonic antigen. In bold are indicated *p* < 0.05.

**Table 3 cancers-14-05921-t003:** Cox models for PFS/OS stratifying by the annotated mutated genes (individually or in combination).

Mutated Genes		PFS				OS			
		Univariate Analysis		Multivariable Analysis		Univariate Analysis		Multivariable Analysis	
		HR (95% CI)	*p*	HR (95% CI)	*p*	HR (95% CI)	*p*	HR (95% CI)	*p*
*TP53*	Mut vs. wt	1.07 (0.77–1.48)	0.70	1.31 (0.84–2.04)	0.24	1.09 (0.71–1.66)	0.69	1.10 (0.65–1.86)	0.72
*RAS*	Mut vs. wt	1.15 (0.84–1.57)	0.40	1.37 (0.92–2.05)	0.12	1.20 (0.80–178)	0.38	0.99 (0.60–1.64)	0.98
*PIK3CA*	Mut vs. wt	1.74 (1.12–2.70)	**0.014**	1.42 (0.79–2.54)	0.24	2.03 (1.21–3.41)	**0.007**	1.53 (0.75–3.13)	0.25
*SMAD4*	Mut vs. wt	1.52 (0.98–2.36)	0.06	2.63 (1.45–4.79)	**0.0015**	1.39 (0.81–2.39)	0.23	1.74 (0.80–3.76)	0.16
*BRAF*	Mut vs. wt	0.93 (0.54–1.58)	0.78	0.87 (0.45–1.65)	0.66	1.54 (0.84–2.79)	0.16	1.40 (0.65–3.03)	0.39
*FBXW7*	Mut vs. wt	1.54 (0.90–2.64)	0.12	1.69 (0.87–3.26)	0.12	1.85 (1–3.41)	**0.049**	2.24 (1.06–4.76)	**0.036**
*TP53*/*RAS*	Double mut vs. *TP53* mut	1.01 (0.69–1.50)	0.94	1.36 (0.81–2.29)	0.25	1.28 (0.79–2.09)	0.32	1.08 (0.57–2.03)	0.82
*TP53*/*PIK3CA*	Double mut vs. *TP53* mut	1.58 (0.82–3.07)	0.17	0.90 (0.32–2.57)	0.85	2.78 (1.35–5.71)	**0.006**	1.17 (9.32–4.23)	0.81
*TP53*/*SMAD4*	Double mut vs. *TP53* mut	1.94 (1.12–3.38)	**0.019**	4.32 (2–9.30)	**0.0002**	1.80 (0.91–3.55)	0.09	2.91 (1.08–7.85)	**0.035**
*TP53*/*FBXW7*	Double mut vs. *TP53* mut	1.72 (0.92–3.24)	0.09	2.65 (1.13–6.23)	**0.025**	1.83 (0.90–3.73)	0.09	3.31 (1.22–8.96)	**0.019**
*TP53*/*BRAF*	Double mut vs. *TP53* mut	2.24 (1.19–4.20)	**0.01**	1.56 (0.71–3.42)	0.27	3.18 (1.56–6.50)	**0.0015**	1.82 (0.63–5.27)	0.27
*RAS*/*PIK3CA*	Double mut vs. *RAS* mut	1.58 (0.89–2.79)	0.12	0.75 (0.28–1.99)	0.56	1.82 (0.63–5.27)	0.20	0.51 (0.13–1.98)	0.33

PFS, progression–free survival; OS, overall survival; HR, hazard ratio; CI, confidence interval; Mut, mutated; wt, wild–type. In bold are indicated *p* < 0.05.

## Data Availability

The data that support the findings of this study are available from the corresponding author upon reasonable request.

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
