# Peer review of "Mutational Status of SMAD4 and FBXW7 Affects Clinical Outcome in TP53–Mutated Metastatic Colorectal Cancer"

_cancers, 2022, doi:10.3390/cancers14235921_

Round 1

Reviewer 1 Report

Major points

1) Table 1; Only 30% of patients received antibody combination therapy. This is very different from the current clinical practice.

2) Table 2; Use of antibody is not related to PFS and OS. This is different from published data.

3) Figure 2; The double mut group is small, 17 cases and 16 cases, respectively. Since there is a possibility that there is a correlation between genetic mutations and response to antibody therapy, an increase in the antibody combination therapy population may have a significant impact on the survival curve of the double mut group.

Reviewer 2 Report

The article is focused on the results obtained from the molecular testing by targeted gene sequencing of metastatic colorectal cancer patients.

The article is interesting, although not easily to be understood from the general audience, in some parts mainly due to the great amount of data presented.

Methodology is properly described and DNA quality was assessed by Qubit. I have concerns about grouping rectal cancer with descendent colon and sigma since generally tumors of the rectum have different features. It might be interesting to compare the different molecular profiles according to tumor location.

The conclusions are consistent with the evidence and arguments
presented and address the main question of the study.

Figures are of good quality and report all significative data, as well as tables.

The reference section is appropriate.

The quality of English is fine.

Reviewer 3 Report

In the manuscript by Lahoz et al. have defined that SMAD4 and FBXW7 mutations can act as negative prognostic markers in TP53-driven colorectal tumors. The author has performed next-generation sequencing in 294-colorectal patient tumor samples. In addition to correlation between SMAD4 and FBXW7 to TP53, the author has further illustrated pathways that are related to SMAD4 and FBXW7 and suggested that these two molecules can be used as biomarkers to predict the prognostic in colorectal cancer patients. The manuscript is well written and well described; the manuscript can be accepted in the present form.

Author Response

We highly appreciate the Reviewer comments and we are very pleased to read that the Reviewer considers that our manuscript can be published in its present form.